# ATROUS LEARNING FOR DIFFUSION MODELS

## ABSTRACT

Diffusion models have shown remarkable success across a wide range of generative tasks. However, they often suffer from spatially inconsistent generation, arguably due to the inherent locality of their denoising mechanisms. For example, a diffusion model trained on natural images might generate hands with six fingers. To mitigate this issue, we propose atrous learning for diffusion models, a simple yet effective masking strategy that can be implemented with only a few lines of code. Experiments show that it is surprisingly safe to mask up to 98% of pixels for diffusion model training. Our method attains competitive FID scores across datasets and avoids training instability on small datasets. Moreover, the masking strategy reduces memorization and promotes the use of broader contextual information during generation.

## 1 INTRODUCTION

Generative modeling aims to approximate and sample from typically unknown data distributions (Albergo et al., 2023). Among the various frameworks proposed (Goodfellow et al., 2014; Kingma & Welling, 2013; Van Den Oord et al., 2016; Papamakarios et al., 2021), diffusion models have achieved remarkable success across diverse domains (Ma et al., 2024; Rombach et al., 2022; Saharia et al., 2022; Blattmann et al., 2023; Kong et al., 2020), largely attributable to their simple regression-based training objective such as to regress the injected noise or score function (Sohl-Dickstein et al., 2015; Song & Ermon, 2019; Ho et al., 2020; Song et al., 2020). A diffusion model progressively perturbs data into Gaussian noise through an iterative stochastic process and then learns to reverse this corruption via a *denoiser*. Consequently, a new Gaussian noise sample can be transformed back into the data distribution via the learned reverse process. More recently, flow matching has unified diffusion models within the framework of probability flows and simplified the generative process by replacing the iterative stochastic dynamics with a straight flow (Lipman et al., 2022; Liu et al., 2022; Albergo & Vanden-Eijnden, 2022; Albergo et al., 2023).

Recent works have been putting effort in understanding how diffusion models convert their training data into novel outputs deviated from the training samples. It is shown that models that learn ideal score functions can only generate memorized training examples (Kamb & Ganguli, 2025; Biroli et al., 2024; Gu et al., 2023; Somepalli et al., 2023), while practical diffusion models necessarily deviate from the ideal denoiser at intermediate timesteps. Among the inductive biases that introduce approximation errors relative to the optimal denoiser (Kadkhodaie et al., 2024; Niedoba et al., 2025; Kamb & Ganguli, 2025), locality has been identified as a key mechanism underpinning the remarkable generalization ability of diffusion models. Niedoba et al. (2025) demonstrate that the behavior of a full image-based denoiser can be replicated by aggregating patch-based local empirical denoisers. However, excessive reliance on locality can also lead to the notorious spatially inconsistent image generation problem (Kamb & Ganguli, 2025; Shen et al., 2024; Lin et al., 2024).

In this paper, motivated by the limitations imposed by locality, we propose *atrous learning* to enhance contextual representations in diffusion models. The term draws inspiration from *atrous convolution* (Chen et al., 2017), however, rather than inserting holes into convolutional kernels, we introduce them into the training losses. Our method, termed *Simplified Masked Diffusion* (SMD), applies random masks to pixel positions when computing the regression loss, notably distinguished from prior masked diffusion approaches designed for discrete spaces (Austin et al., 2021; Shi et al., 2024). As a result, the model learns solely from unmasked pixels while being encouraged to generalize over masked regions. This approach is easy to implement and exhibits surprisingly strong

performance, even when up to 98% of pixels are masked. Additionally, it mitigates memorization (Hans et al., 2024) in diffusion models and helps prevent training divergence.

## 2 RELATED WORK

We begin by reviewing diffusion models and outlining the spatial inconsistency problem that motivates our study. Next, we examine mask modeling across various applications. Finally, we focus on works that incorporate masking techniques within diffusion models.

**Diffusion Models** Despite the remarkable success of diffusion (Ho et al., 2020; Song et al., 2020) and flow matching models (Lipman et al., 2022; Albergo et al., 2023) in image and video generation (Rombach et al., 2022; Ma et al., 2024), their theoretical underpinnings remain insufficiently understood, particularly with respect to their surprising generalization capabilities. Recent studies suggest that the empirical optimal denoiser is only capable of reproducing training samples (Biroli et al., 2024; Gu et al., 2023), in contrast to the novel generations observed in practice. Nonetheless, memorization effects can emerge when the training dataset is small (Kadkhodaie et al., 2024). Several works have investigated the inductive biases inherent in diffusion models that give rise to such novel generations (Kadkhodaie et al., 2024; Kamb & Ganguli, 2025; Niedoba et al., 2025), among which locality has been identified as a key mechanism. Specifically, locality enables diffusion models to deviate from strictly learning the optimal denoiser, thereby allowing them to generate unseen samples. However, locality has also been recognized as the primary cause of spatial inconsistency in generation (Kamb & Ganguli, 2025). In this paper, we address this limitation by encouraging diffusion models to capture broad contextual representations, thereby mitigating the inconsistency induced by locality.

**Mask Modeling** Mask modeling has proven effective for both representation learning and generation in language and vision domains. In natural language processing (NLP), transformer-based models (Vaswani et al., 2017) trained on next-token prediction or masked-token prediction objectives exhibit strong generalization in large-scale pretraining (Devlin et al., 2019; Song et al., 2019) and language generation (Radford et al., 2019; Brown et al., 2020). Similar strategies have been successfully applied in computer vision, where mask modeling has taken the form of denoising corrupted pixels (Vincent et al., 2010), inpainting (Pathak et al., 2016), or autoregressive prediction (Chen et al., 2020). Inspired by advances in NLP, recent visual representation learning approaches employ transformers to predict masked pixels (Chen et al., 2020), patches (Dosovitskiy et al., 2020; He et al., 2022), or discrete tokens (Zhou et al., 2021). Visual mask modeling has also been extended to generative tasks. Mask Generative Models (MGM), such as MaskGIT, leverage masked transformers to predict masked image tokens for generation (Chang et al., 2022; 2023), with subsequent extensions into continuous spaces (Tschannen et al., 2024; Li et al., 2024). In this work, however, we are interested in how masking can improve diffusion models, which is orthogonal to these prior directions.

**Masking in Diffusion Models** Recent discrete diffusion models incorporate masking as a replacement for Gaussian noise in continuous spaces, primarily to adapt diffusion to discrete domains such as text and code (Austin et al., 2021; Shi et al., 2024; Gat et al., 2024). Our motivation differs from these approaches as we still focus on continuous spaces. The most relevant work is the Masked Diffusion Transformer (MDT) (Gao et al., 2023), which exposes the model only to unmasked patches and trains it to predict the missing ones. However, MDT relies on an asymmetric encoder–decoder design, akin to Masked Autoencoders (MAE) (He et al., 2022), limiting its applicability to general diffusion frameworks. In contrast, our method introduces masking directly into the regression loss, making it architecture-agnostic and straightforward to implement.

## 3 BACKGROUND

We provide an overview of diffusion models and flow matching models, which are mathematically equivalent (Albergo et al., 2023). In particular, we introduce the concept of the optimal denoiser and discuss how the empirical denoiser tends to memorize training samples.

## 3.1 DIFFUSION MODELS

Given an unknown data distribution, instead of directly estimating the probability density $p(\boldsymbol{x}), \boldsymbol{x} \in \mathbb{R}^d$, diffusion models learn the score function (Song et al., 2020) or denoiser (Ho et al., 2020) from noise-corrupted data to iteratively transform noises from a prior distribution to the target data distribution.

**Forward Process** The forward process of diffusion models can be described via stochastic differential equations (SDE):

$$\mathrm{d}\boldsymbol{z} = f(\boldsymbol{z}, t)\,\mathrm{d}t + g(t)\,\mathrm{d}\boldsymbol{w}, \tag{1}$$

where $\boldsymbol{z} \in \mathbb{R}^d$ represents intermediate corrupted samples during the diffusion process, $f(\boldsymbol{z}, t)$ and $g(t)$ are known as the drift and diffusion functions. $\boldsymbol{w}(t)$ is the standard Wiener process. For each timestep $t \in (0, T]$, we obtain the marginal distribution $p_t(\boldsymbol{z}) = \int p_t(\boldsymbol{z} \mid \boldsymbol{x})p(\boldsymbol{x})\,\mathrm{d}\boldsymbol{x}$ from Eq. (1). Generally, by setting proper $f(\boldsymbol{z}, t)$ and $g(t)$, we would like to have $p_t(\boldsymbol{z} \mid \boldsymbol{x})$ as a Gaussian distribution with closed-form mean and variance.

**Backward Process** The core objective of diffusion models is to learn the time-reversal of Eq. (1). This reverse process is governed by the corresponding reverse-time SDE (Song et al., 2020):

$$\mathrm{d}\boldsymbol{z} = \left[f(\boldsymbol{z}, t) - g(t)^2 \nabla_{\boldsymbol{z}} \log p_t(\boldsymbol{z})\right] \mathrm{d}t + g(t)\,\mathrm{d}\tilde{\boldsymbol{w}}. \tag{2}$$

Karras et al. (2022) demonstrate that multiple parameterizations of $f(\boldsymbol{z}, t)$ and $g(t)$ are equivalent. By adopting the parameterization with $f(\boldsymbol{z}, t) = 0$ and $g(t) = \sqrt{2t}$, it yields transition distributions $p_t(\boldsymbol{z} \mid \boldsymbol{x}) = \mathcal{N}(\boldsymbol{x}, t^2 \boldsymbol{I}_d)$ and prior $\pi(z) = \mathcal{N}(\boldsymbol{0}, T^2 \boldsymbol{I}_d)$. Note that the standard deviation of added noise here is $\sigma(t) = t$.

The reverse SDE in Eq. (2) requires estimation of the score function $\nabla_{\boldsymbol{z}} \log p_t(\boldsymbol{z})$. For the chosen diffusion process, the score function takes the explicit form (Niedoba et al., 2025):

$$\nabla_{\boldsymbol{z}} \log p_t(\boldsymbol{z}) = \frac{\mathbb{E}[\boldsymbol{x} \mid \boldsymbol{z}, t] - \boldsymbol{z}}{t^2}. \tag{3}$$

Note that the score estimation in Eq. (3) is mathematically equivalent to estimating the posterior mean $\mathbb{E}[\boldsymbol{x} \mid \boldsymbol{z}, t]$, which is also the objective of denoising. Since the true data distribution $p(\boldsymbol{x})$ is generally unknown, exact computation of the posterior $p_t(\boldsymbol{x} \mid \boldsymbol{z})$ and hence $\mathbb{E}[\boldsymbol{x} \mid \boldsymbol{z}, t]$ is intractable. Instead, diffusion models employ neural networks as denoisers to approximate $\mathbb{E}[\boldsymbol{x} \mid \boldsymbol{z}, t]$. These networks are trained using an empirical data distribution $p_{\mathcal{D}}(\boldsymbol{x}) = \frac{1}{N} \sum_{\boldsymbol{x}^{(i)} \in \mathcal{D}} \delta(\boldsymbol{x} - \boldsymbol{x}^i)$, where the dat aset is $\mathcal{D} = \{\boldsymbol{x}^1, \ldots, \boldsymbol{x}^N \mid \boldsymbol{x}^i \sim p(\boldsymbol{x})\}$. The training objective is:

$$\mathbb{E}_{\boldsymbol{x}^i \sim p_{\mathcal{D}}(\boldsymbol{x}), \boldsymbol{z} \sim p_t(\boldsymbol{z} \mid \boldsymbol{x}^i), t \sim p(t)} \left[\lambda(t) \left\|\boldsymbol{x}^i - D_\theta(\boldsymbol{z}, t)\right\|^2\right], \tag{4}$$

where $\lambda(t)$ is a weighting function and $D_\theta(\boldsymbol{z}, t)$ represents the neural network denoiser.

**Optimal Denoiser** The theoretical minimizer of Eq. (4) and the optimal denoiser for any $(z, t)$ pair is the empirical posterior mean (Vincent et al., 2010; Karras et al., 2019):

$$\mathbb{E}_{\boldsymbol{x} \sim p_D}[\boldsymbol{x} \mid \boldsymbol{z}, t] = \sum_{\boldsymbol{x}^i \in D} p_t(\boldsymbol{x}^i \mid \boldsymbol{z})\boldsymbol{x}^i, \tag{5}$$

which is the average over the images of the data set $\mathcal{D}$, weighted by their posterior probability. Note that the empirical optimal denoiser in Eq. (5) can only generate samples in the training dataset $\mathcal{D}$, i.e., *the optimal denoiser memorizes* (Kamb & Ganguli, 2025). How Eqs. (2) and (3) result in *memorization* using the optimal denoiser in Eq. (5) has been well established in Biroli et al. (2024).

## 3.2 FLOW MATCHING

Flow matching provides a unified perspective on diffusion models by directly learning a time-dependent vector field that transforms noisy data $z$ into the target data distribution. This transformation admits multiple probability paths, such as diffusion paths and linear conditional paths,

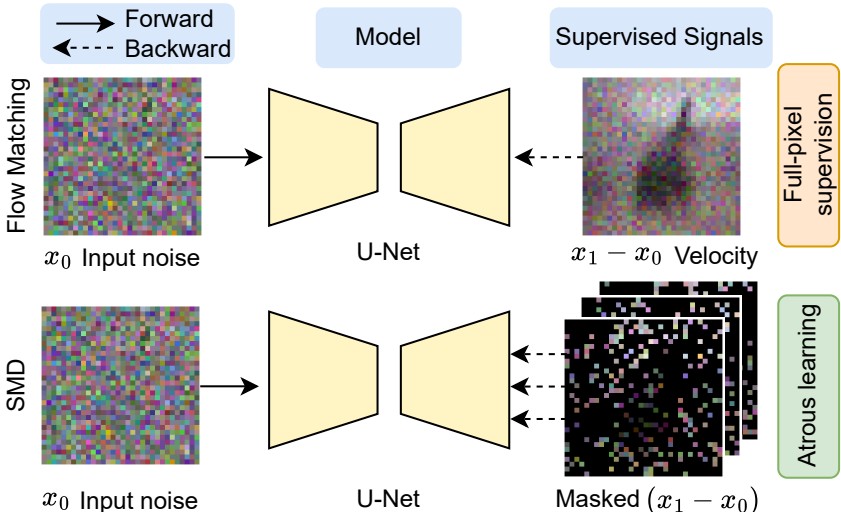

Figure 1: Overview of the proposed method in comparison with standard flow matching (FM). The proposed SMD differs from FM via *atrous learning*, which uses masked supervised signals to prevent diffusion models from memorizing training data points and to encourage the model to leverage contextual information when predicting neighboring pixels.

with diffusion models arising as a special case. In this section, we focus on the objective of learning a linear conditional probability path:

$$\mathcal{L}_{\text{CFM}} = \mathbb{E}_{t,\pi(\boldsymbol{x}_0),p(\boldsymbol{x}_1)} \left\| \boldsymbol{v}_\theta(t, \boldsymbol{z} \mid \boldsymbol{x}_1) - (\boldsymbol{x}_1 - \boldsymbol{x}_0) \right\|^2, \qquad (6)$$

where $\boldsymbol{x}_0 \in \pi(\boldsymbol{x}_0)$ denotes noise samples from a prior distribution, $\boldsymbol{x}_1 \in p(\boldsymbol{x}_1)$ are target data samples, and $\boldsymbol{z}$ is the intermediate state. By learning the vector field $\boldsymbol{v}_\theta(t, \boldsymbol{z})$, flow matching iteratively transforms noise samples into data samples through an ODE process

$$\boldsymbol{z}_{t+h} = \boldsymbol{z}_t + h\boldsymbol{v}_\theta(\boldsymbol{z}_t, t), \qquad (7)$$

where $h > 0$ is a user-defined time step.

## 4 METHODS

We first show that practical diffusion models deviate from the optimal denoiser due to their inherent locality (Kamb & Ganguli, 2025; Niedoba et al., 2025; Kadkhodaie et al., 2024). Building on this observation, we introduce simplified masked diffusion to encourage the model to capture global contextual representations. We further analytically show that our method provides an unbiased gradient estimator of standard diffusion models while introducing higher variance to enhance exploration.

### 4.1 LOCALITY IN DIFFUSION MODELS

Inductive biases embedded in practical diffusion models (Ho et al., 2020; Peebles & Xie, 2023) have been shown helping the models deviate from the optimal denoiser. One of notable inductive bias is identified as *locality* (Kamb & Ganguli, 2025):

**Definition 1** (Ω-locality). *A model $\mathcal{M}_t(\boldsymbol{x})$ based on image $\boldsymbol{x}$, is defined to be Ω-local if, for all images $\boldsymbol{x}$ and all pixel locations $j$, $\mathcal{M}_t(\boldsymbol{x})[j]$ depends on $\boldsymbol{x}$ only through $\boldsymbol{x}_{\Omega_j}$, i.e., $\mathcal{M}_t(\boldsymbol{x})[j] = \mathcal{M}_t(\boldsymbol{x}_{\Omega_j})[j]$, where $\Omega_j$ denotes a local neighborhood of pixel position $[j]$ in image $\boldsymbol{x}$ and $\boldsymbol{x}_{\Omega_j}$ represents the pixel values of $x$ in the area $\Omega_j$.*

The *locality* has also been empirically verified in Niedoba et al. (2025), which proposes to use a set of patch-based denoisers to approximate the full-image based diffusion network denoisers. With proper batch set design, patch-based models can well approximate the full-image based denoiser,

which means that diffusion models do not necessarily use global information to generate new samples. However, we hypothesize that encouraging diffusion models to exploit broader contextual information can lead to the generation of more realistic samples.

## 4.2 SIMPLIFIED MASKED DIFFUSION

To address the above limitation, we introduce *Simplified Masked Diffusion* (SMD), a method designed to mitigate the locality inherent in current diffusion models and to promote the learning of global contextual representations. The masking strategy introduced in SMD is general and can be applied to both diffusion models and flow-matching models. For clarity of exposition, we adopt a unified regression objective to illustrate the mechanism of SMD:

$$\mathcal{L}_{\text{SMD}}(f) = \mathbb{E}_{\boldsymbol{x} \sim p(\boldsymbol{x})} \mathbb{E}_{M \sim q(M)} \Big[ \sum_{j:M_j=1} \|f(\boldsymbol{x})[j] - v^*(\boldsymbol{x})[j]\|^2 \Big], \tag{8}$$

where $M \in \{0,1\}^d$ denotes a randomly sampled binary mask, with each entry $M_j \sim \text{Bernoulli}(1-\eta)$, and $d$ is the dimensionality of the image $\boldsymbol{x}$. The mask ratio $\eta \in (0,1)$ is used to control the proportion of visible elements, enabling us to analyze the impact of SMD on performance. In denoising diffusion models, $f(\boldsymbol{x})$ and $v^*(\boldsymbol{x})$ correspond to the denoiser network output and the target image, respectively, whereas in flow-matching models they represent the learned velocity field and the target conditional vector field. Through masking, the objective $\mathcal{L}(f)$ is optimized only over the unmasked pixel positions, i.e., $f_j(\boldsymbol{x})$ and $v_j^*(\boldsymbol{x})$ with $M_j = 1$.

**Remark 1.** *The core idea of atrous learning is by masking pixels in the target signals, we enforce the model to utilize broader contextual representation to generate full pixels, as shown in Fig. 1.*

## 4.3 SMD IS AN UNBIASED GRADIENT ESTIMATOR BUT WITH HIGHER VARIANCE

Since the masks are sampled randomly and independently of the images, it can be readily shown that the SMD objective preserves the gradient direction in an unbiased manner, as in standard diffusion models. At the same time, the masking strategy increases the variance of the gradients, thereby facilitating exploration during learning.

**Proposition 1.** *Let the mask ratio be $\eta \in (0,1)$, i.e., each pixel is masked independently with probability $\eta$, Then, SMD provides an unbiased estimate of the gradient direction, as in standard diffusion models, while increasing the gradient variance by a factor of $\frac{\eta}{1-\eta}$.*

*Proof.* Let $e_j(\boldsymbol{x}) = \|f_\theta(\boldsymbol{x})[j] - v^*(\boldsymbol{x})[j]\|^2$ and define the normal objective without masking as

$$\mathcal{L}_{\text{normal}}(f) = \mathbb{E}_{\boldsymbol{x} \sim p(\boldsymbol{x})} \sum_{j=1}^{d} e_j(\boldsymbol{x}). \tag{9}$$

Let $M \in \{0,1\}^d$ be a random mask with independent entries $M_j \sim \text{Bernoulli}(p)$, where $p = 1 - \eta \in (0,1)$. Note $\boldsymbol{g}_j := \nabla_\theta e_j(\boldsymbol{x}) \in \mathbb{R}^m$ for a fixed $\boldsymbol{x}$. Then for any fixed image $\boldsymbol{x}$ and mask $M$, the masked objective and gradient w.r.t. the model parameter $\theta$ are

$$\mathcal{L}_{\text{SMD}}(f; M, \boldsymbol{x}) = \sum_{j=1}^{d} M_j \, e_j(\boldsymbol{x}), \quad g_{\text{SMD}}(M, \boldsymbol{x}) = \sum_{j}^{d} M_j \boldsymbol{g}_j. \tag{10}$$

Since expectation is linear and $\mathbb{E}[M_j] = p$, it is easy to see $g_{\text{SMD}}$ is an unbiased estimation of $g_{\text{normal}}$ with a constant factor $p$:

$$\mathbb{E}_{\boldsymbol{x},M}\big[\mathcal{L}_{\text{SMD}}(f; M, \boldsymbol{x})\big] = p \cdot \mathcal{L}_{\text{normal}}(f), \ \mathbb{E}_M\big[\boldsymbol{g}_{\text{SMD}(M,\boldsymbol{x})}\big] = p \cdot \sum_{j}^{d} \nabla_\theta e_j(\boldsymbol{x}) = p \cdot \boldsymbol{g}_j(\boldsymbol{x}). \tag{11}$$

Because $\text{Var}(M_i) = p(1-p)$, the covariance of $\boldsymbol{g}_{\text{SMD}}$ is

$$\text{Cov}[\boldsymbol{g}_{\text{SMD}}] = \sum_{j=1}^{d} \text{Var}(M_j)\boldsymbol{g}_j\boldsymbol{g}_j^\top = \sum_{j=1}^{d} p(1-p) \, \boldsymbol{g}_j\boldsymbol{g}_j^\top. \tag{12}$$

By scaling the gradient $g_{\text{SMD}}$ with a constant factor $\frac{1}{p}$, we can get an unbiased estimation with covariance

$$\text{Cov}\left[\frac{\boldsymbol{g}_{\text{SMD}}}{p}\right] = \frac{1-p}{p}\sum_{j=1}^{d}\boldsymbol{g}_j\boldsymbol{g}_j^{\top}, \tag{13}$$

which inflates the gradient variance as $p$ decreases, i.e. when mask ratio $\eta$ increases.

$\square$

## 5 EXPERIMENTS

We design a series of experiments to answer the following research questions: (1) Does the proposed atrous learning improve the spatial consistency of generated samples? (2) Does it enable the model to leverage broader contextual information during generation? (3) Does the masking strategy affect FID scores? (4) Can the method assist in estimating the population score function, the fundamental objective of generative models? (5) Does it help stabilize training on small datasets? (6) Can it mitigate memorization in diffusion-based generation?

### 5.1 VISUALIZE SPATIAL CONSISTENCY

In this experiment, we construct a toy dataset comprising 500 binary images containing randomly positioned squares and triangles. We compare the proposed SMD approach with the original flow matching (FM) implementation from Lipman et al. (2024). Fig. 2 presents generated samples from SMD ($\eta = 0.5$) and FM ($\eta = 0.0$), alongside samples from the training dataset. The results show that, with masking, SMD produces fewer unstructured shapes, whereas the baseline FM model occasionally generates scattered dots in some images.

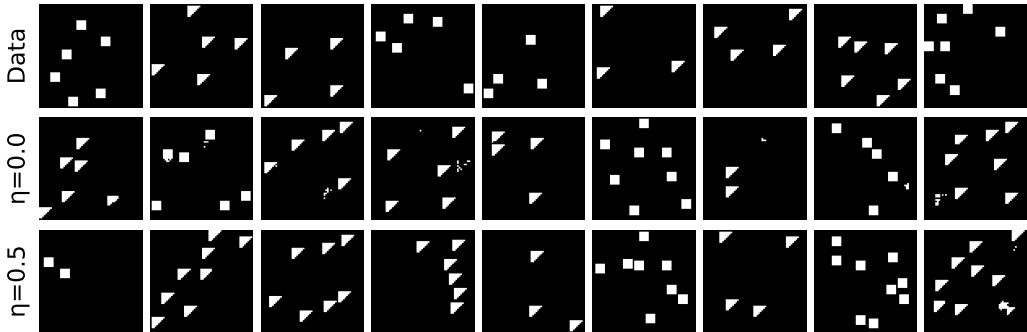

Figure 2: After training for $5,000$ epochs on a dataset of $500$ images, our model generates images with enhanced structural integrity compared to the standard diffusion model without masking.

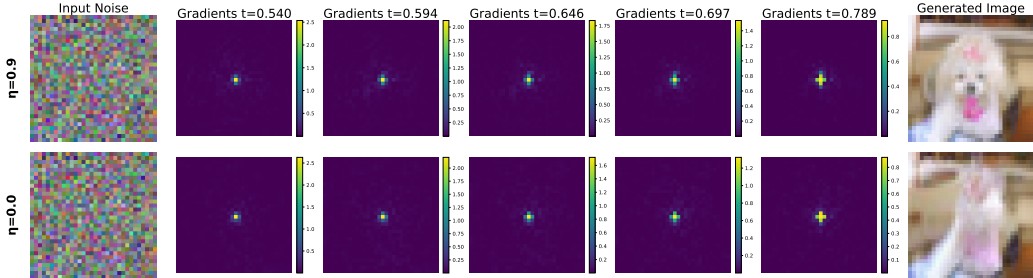

Figure 3: Gradient sensitivity heatmaps computed from models with $\eta = 0.9$ and $\eta = 0$.

## 5.2 SMD PROMOTES BROADER CONTEXTUAL REPRESENTATION

To evaluate the contextual representation learned by models, we adopt the gradient sensitivity maps used in Niedoba et al. (2025) to measure how the pixel positions are correlated in different models. For each timestep $t$, the gradient sensitivity heatmap is defined as

$$G(x,y,t) = \mathbb{E}_{z \sim p_t(x^{(i)},z)} \left[ \sum_{c=1}^{3} |\nabla_{z_c} v_\theta(z,t)_{x,y,c}| \right], \qquad (14)$$

where $v_\theta(z,t)_{x,y,c}$ denotes the output of the vector field network at pixel position $(x,y)$ and image channel $c$. In our experiments, we compute gradients at the central point of each image. The higher gradient usually means a stronger influence from contextual pixels when generating a pixel.

**Visualizing Gradient Sensitivity**  As shown in Fig. 3, although both models are trained on CIFAR10 and receive identical noise inputs, they eventually generate different images, which is contrary to the typical observation that diffusion models produce similar outputs under shared noise (Kadkhodaie et al., 2024). This divergence arises from differences in their gradients at critical timesteps where mode selection occurs. At $t = 0.540$, the gradients remain similar, but as $t$ increases, the model with a 90% mask ratio exhibits broader gradient sensitivity, indicated by brighter pixels around the center. This broader sensitivity allows our model to form a clearer, dog-like structure, while the baseline collapses into an unidentifiable object.

**Quantified Gradient Sensitivity**  We further compute the average gradient sensitivity across 10,000 images. Specifically, we evaluate the L1 norm of gradients, $\|G(x,y,t)\|_1$, where $x,y$ denotes the central pixel locations of the images, and $t = 0.789$ corresponds to a lower noise level. Fig. 4 shows a clear distribution shift when $\eta = 0.9$. Compared with the baseline ($\eta = 0$), SMD exhibits substantially larger gradients, indicating that the model generates pixels using broader and stronger contextual information.

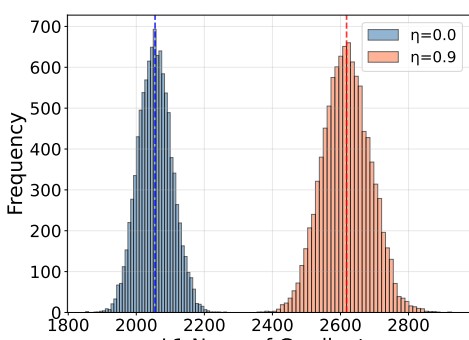

Figure 4: Distribution of L1 norm of gradients over 10,000 images.

## 5.3 FID ON LARGE DATASETS

**Setup**  We evaluate the method on four large-scale datasets, CIFAR10 32×32, CelebA-50K 64×64, LSUN Bedroom 32×32 and ImageNet 32×32. Due to the high resolution of CelebA, we only use 50,000 samples from the dataset. For all methods, we use the same convolution U-Net model (Ronneberger et al., 2015) and the number of function evaluations (NFE) is fixed at 50. We employ Heun's second-order method (Ascher & Petzold, 1998) as the ODE solver, with the sampling strategy proposed by Karras et al. (2022). We compute the Fréchet Inception Distance (FID) (Heusel et al., 2017) with 50,000 generated samples.

**Quantitative Results**  We plot the evaluation FID scores across training epochs to visualize the learning dynamics, with curves averaged over four runs. As shown in Fig. 5, models with an 80% masking ratio generally achieve FID performance comparable to the unmasked baseline. Notably, in Fig. 5b, the baseline model diverges after 1,500 epochs, whereas our method with $\eta = 0.8$ remains stable. This phenomenon is further validated in our CelebA-10K experiments, where the baseline exhibits even more severe divergence. More results can be found in Section F.

**Qualitative Results**  For qualitative comparison, we visualize generated samples from different models in Fig. 6. The same noises are fed into each model to observe how the generated images change with varying mask ratios $\eta$. While the images generated from the same noise generally appear similar, we observe improved spatial consistency as $\eta$ increases. For instance, the unrealistic bulge on the head in sample #6 gradually diminishes as $\eta$ increases. Moreover, sample #1 shows more detailed neck wrinkles with $\eta = 0.8$ compared to the image with $\eta = 0$.

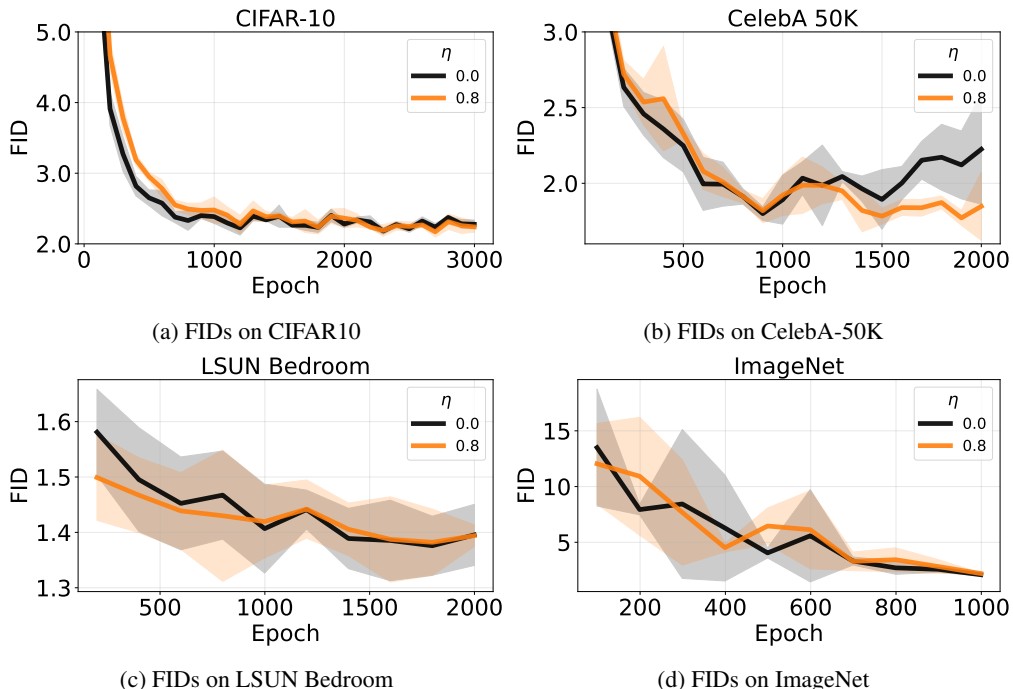

(a) FIDs on CIFAR10

(b) FIDs on CelebA-50K

(c) FIDs on LSUN Bedroom

(d) FIDs on ImageNet

Figure 5: Comparison of evaluation FIDs during training. When $\eta = 0$, the model reduces to the baseline flow matching (Lipman et al., 2024). Across four datasets, SMD with up to 80% masked pixels can still achieve comparable performance as the baseline. Notably, the baseline in Fig. 5b eventually explodes while SMD remains stable. The shaded region indicates the 95% confidence interval over four runs.

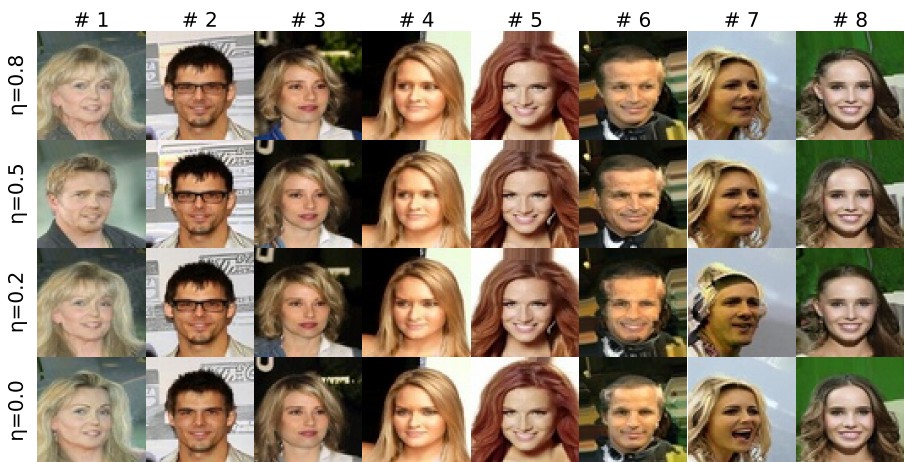

Figure 6: Visualization of non-curated generated samples from models trained on CelebA dataset. Sample #1 shows more detailed neck wrinkles with $\eta = 0.8$ compared to $\eta = 0$. In sample #6, an unrealistic bulge on the head is present at $\eta = 0$ but gradually disappears as $\eta$ increases.

## 5.4 SCORE FUNCTION APPROXIMATION ERROR

FID is limited in its ability to assess how closely generated samples match the true data distribution. This limitation helps explain why diffusion models with excellent FID scores can still produce unrealistic images, even when trained exclusively on natural images (Bonnaire et al., 2025). Fundamentally, this issue arises because the training objective estimates only the empirical score function

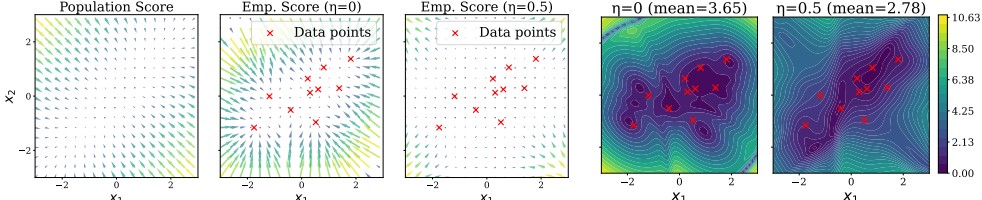

Figure 7: Visualization of population scores (ground truth) and empirical scores estimated with $\eta = 0$ and $\eta = 0.5$, along with a comparison of score estimation errors (right two figures). The baseline model only accurately estimates scores near observed data points, whereas masking estimation allows estimation across broader regions and achieves lower estimation errors.

rather than the true underlying score function, thereby leading to both memorization and unrealistic generations.

**Score Estimation Comparison** Since we generally do not know the ground truth score function of real datasets (Ho et al., 2020), we design experiments on synthesized 2D Gaussian data, where we can analytically compute the score function. We compare the approximated empirical score functions obtained by the baseline and our method under different masking ratios. Fig. 7 visualizes the estimated scores on a grid over the square, based on the given ten data points. By applying masking ($\eta = 0.5$), which generates multiple partial views of the data points, the estimation becomes more accurate over a broader region. The two figures on the right in Fig. 7 presents the estimation errors, defined as the absolute difference from the population score, with and without masking. The masked estimation achieves an average error of 2.78, which is lower than the baseline's average error of 3.65. The detailed description of the numerical experiment is in Section B.

### 5.5 MASKING MITIGATES TRAINING DIVERGENCE

We further compare SMD with the baseline on CelebA-10K at 64×64 resolution, which contains 10,000 images. As shown in Fig. 8, SMD performs well even with masking ratios as high as 98%, and it effectively mitigates the severe divergence observed in baseline models trained for long time without masking or with lower masking ratios such as $\eta = 0.5$.

### 5.6 MASKING MITIGATES MEMORIZATION

We also observe that the proposed masking strategy can mitigate the memorization problem in diffusion model training, which is particularly critical for small datasets. We generate 10,000 images using models trained on CelebA-10K and

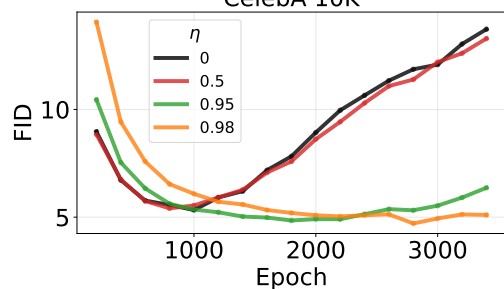

Figure 8: Evaluation FIDs during training on CelebA-10K.

compute the L2 distances to their nearest neighbors in the training set. The results in Table 1 show that, with $\eta = 0.98$, SMD achieves higher L2 distances compared to the baseline without masking, indicating that SMD generates images that are more distinct from the training samples, while maintaining comparable or even lower FID, as illustrated in Fig. 8. The distance computation follows Bonnaire et al. (2025), $d_{\text{mem}} = \|x_\tau - a^\mu\|_2$, where $x_\tau$ is a generated sample and $a^\mu$ is the nearest neighbor of $x_\tau$ in the training dataset.

Table 1: Averaged L2 distance to nearest training samples over 10,000 images.

| Mask ratio $\eta$ | 0.98 | 0.0 |
|---|---|---|
| L2 Distance (↑) | **46.02 (8.03)** | 42.32 (7.85) |

## 6 DISCUSSION AND CONCLUSION

Although diffusion models can generate high-quality images, the issue of spatial inconsistency remains a significant challenge. Motivated by the locality mechanism inherent in diffusion models, we have proposed a simple masking strategy that encourages the model to leverage contextual information when predicting unseen pixel positions (Siméoni et al., 2025). The proposed SMD thus offers a *free lunch* for diffusion model training with several advantages: (1) It achieves comparable FID scores across datasets while avoiding long-training instability. (2) It improves underlying population score estimation. (3) SMD can mitigate the memorization problem in diffusion models. (4) Remarkably, SMD maintains comparable performance even when up to 98% of pixels are masked, suggesting valuable implications for understanding the training dynamics of diffusion models.

### REPRODUCIBILITY STATEMENT

We provide detailed hyperparameters used in our experiments in Section A. Upon acceptance of the paper, we will make a reference code implementation together with the experiments available on GitHub under the MIT license.

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

APPENDICES

## A   IMPLEMENTATION DETAILS

Table 2: Key hyperparameters for experiments on three datasets

| Parameter | CelebA-50K 64×64 | CIFAR10 | LSUN Bedroom 32×32 |
|---|---|---|---|
| *Training Configuration* | | | |
| Epochs | 1400 | 3000 | 2000 |
| Effective Batch Size | 128 | 256 | 1024 |
| Learning Rate | 0.0001 | 0.0001 | 0.0001 |
| Optimizer Betas | [0.9, 0.95] | [0.9, 0.95] | [0.9, 0.95] |
| Use EMA | True | True | True |
| *Flow Matching Configuration* | | | |
| Skewed Timesteps | True | True | True |
| EDM Schedule | True | True | True |
| *Sampling Configuration* | | | |
| ODE Method | Heun2 | Heun2 | Heun2 |
| Number of Function Evaluations | 50 | 50 | 50 |
| *Dataset & Evaluation* | | | |
| Number of Images | 50,000 | 50,000 | 303,125 |
| FID Samples | 50,000 | 50,000 | 50,000 |
| Evaluation Frequency | 100 | 100 | 200 |
| *Model Architecture* | | | |
| Input/Output Channels | 3 | 3 | 3 |
| Model Channels | 128 | 128 | 128 |
| Number of ResBlocks | 4 | 4 | 4 |
| Attention Resolutions | [2, 4] | [2] | [2] |
| Dropout | 0.2 | 0.3 | 0.3 |
| Channel Multipliers | [1, 2, 2, 4] | [2, 2, 2] | [2, 2, 2] |
| Convolution Resample | True | False | False |
| Number of Heads | 2 | 1 | 1 |
| Head Channels | 64 | -1 | -1 |
| Scale Shift Norm | True | True | True |
| ResBlock Up/Down | True | False | False |
| New Attention Order | True | True | True |
| *System Configuration* | | | |
| Number of GPUs | 8 | 8 | 8 |

## B   DESCRIPTION OF 2D GAUSSIAN EXPERIMENT

This section describes the detailed setting of the score estimation experiment in Section 5.4, including the 2D Gaussian distribution we use, how we compute the population score, and how we compute the empirical scores with and without masking.

**2D Gaussian Distribution**   We consider a 2D Gaussian distribution as our data distribution:

$$p_0(x) = \mathcal{N}(x; 0, \Sigma), \tag{15}$$

where $x = (x_1, x_2)^T \in \mathbb{R}^2$ and the covariance matrix is

$$\Sigma = \begin{bmatrix} 1 & \rho \\ \rho & 1 \end{bmatrix}. \tag{16}$$

Here, $\rho \in [-1, 1]$ is the correlation coefficient between the two dimensions: $\rho = 0$ corresponds to independent dimensions, $\rho > 0$ to positive correlation, and $\rho < 0$ to negative correlation. In our

experiment, we set $\rho = 0.7$ to to account for the fact that in many high-dimensional datasets, such as images, the dimensions exhibit strong correlations. The probability density function is

$$p_0(x) = \frac{1}{2\pi\sqrt{1-\rho^2}} \exp\left(-\frac{1}{2(1-\rho^2)}(x_1^2 - 2\rho x_1 x_2 + x_2^2)\right). \tag{17}$$

**Forward Diffusion Process**   Given data $x_0 \sim p_0(x)$, the forward noising process is defined as

$$x_t = e^{-t}x_0 + \sqrt{1-e^{-2t}}\,\epsilon, \quad \epsilon \sim \mathcal{N}(0, I), \tag{18}$$

where $\delta_t = 1 - e^{-2t}$ denotes the noise variance at time $t$. We set $t = 0.1$ to reflect low-level noises.

**Population Score Function**   For a Gaussian distribution with covariance $\Sigma$, the score function at time $t$ is

$$\nabla_x \log p_t(x) = -\Sigma_t^{-1}x, \tag{19}$$

where the time-dependent covariance is

$$\Sigma_t = e^{-2t}\Sigma + \delta_t I. \tag{20}$$

**Empirical Score Estimation**   Given training data $\{x_0^{(i)}\}_{i=1}^n$, the empirical score at a query point $x$ is estimated using kernel density estimation:

$$\nabla_x \log \hat{p}_t(x) = \sum_{i=1}^n w_i(x) \cdot \frac{x_t^{(i)} - x}{\delta_t}, \tag{21}$$

with

$$x_t^{(i)} = e^{-t}x_0^{(i)}, \quad d_i(x) = \frac{\|x_t^{(i)} - x\|^2}{\delta_t}, \quad w_i(x) = \frac{\exp(-\frac{1}{2}d_i(x))}{\sum_{j=1}^n \exp(-\frac{1}{2}d_j(x))}.$$

**Masked Score Estimation**   When a mask $m^{(i)} \in \{0, 1\}^d$ indicates observed dimensions for each sample, the masked score is

$$\nabla_x \log \hat{p}_t^{\text{mask}}(x) = \sum_{i=1}^n w_i^{\text{mask}}(x) \cdot \frac{m^{(i)} \odot (x_t^{(i)} - x)}{\delta_t}, \tag{22}$$

where

$$w_i^{\text{mask}}(x) = \frac{\exp\left(-\frac{1}{2}d_i^{\text{mask}}(x)\right)}{\sum_{j=1}^n \exp\left(-\frac{1}{2}d_j^{\text{mask}}(x)\right)}, \tag{23}$$

and

$$d_i^{\text{mask}}(x) = \frac{\|m^{(i)} \odot (x_t^{(i)} - x)\|^2}{\delta_t}.$$

The final masked score is obtained by averaging over multiple random masks, where each dimension is observed independently with probability $\eta$.

## C   GENERATION PROCESS VISUALIZATION

We visualize the sample paths from the same initial noise using models trained with different mask ratios. The sampling is based on the method proposed in Karras et al. (2022) and we use Heun's second-order method as the ODE solver (Ascher & Petzold, 1998). We set NFE as 50. The sample paths in Fig. 9 show that the models may diverge from around $t = 0.58$.

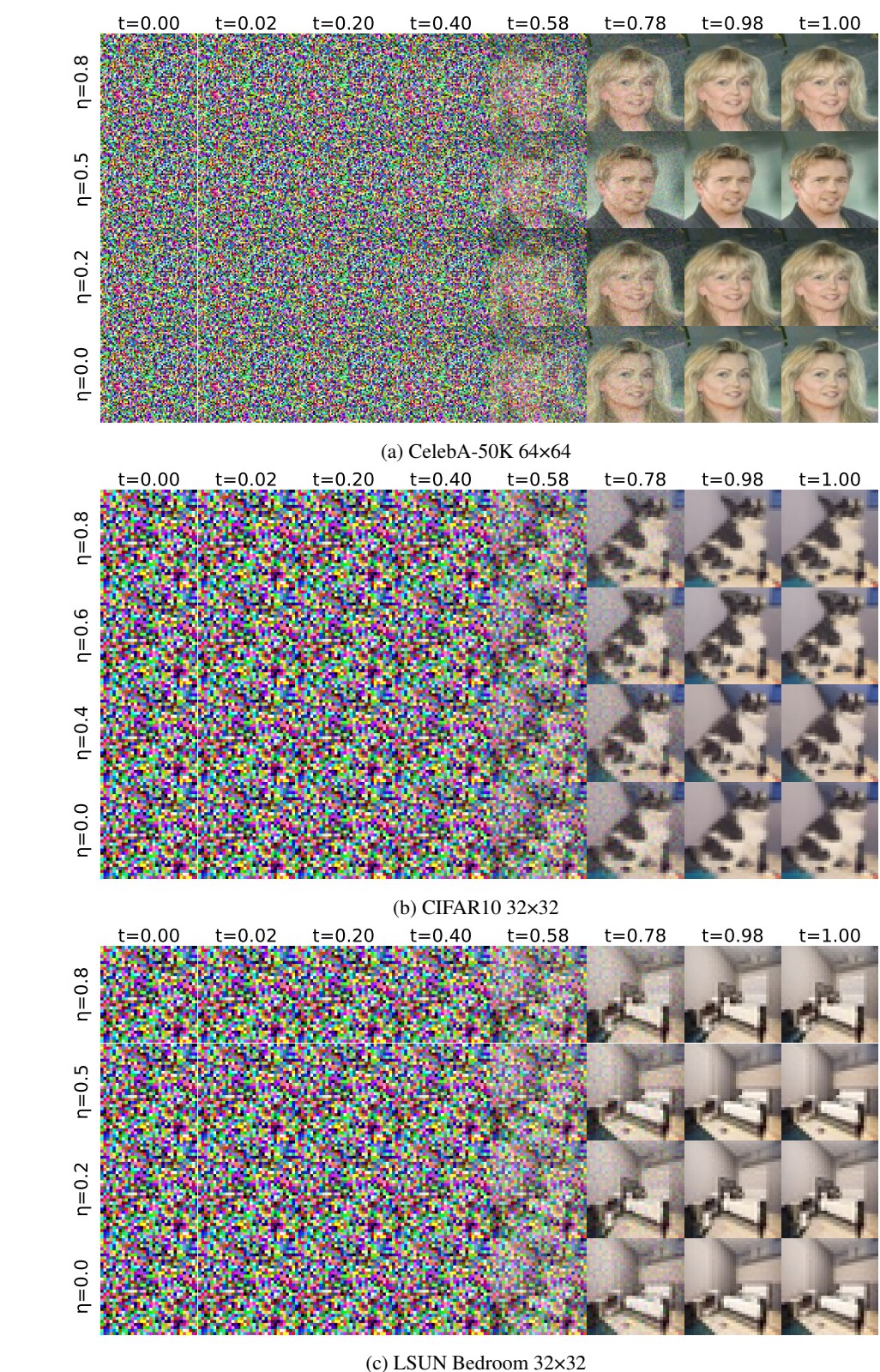

(a) CelebA-50K 64×64

(b) CIFAR10 32×32

(c) LSUN Bedroom 32×32

Figure 9: Sample paths from same initial noises with models trained with different $\eta$ across three datasets.

## D    MORE GENERATED SAMPLES

η=0.8    η=0.5    η=0.2    η=0.0    η=0.8    η=0.5    η=0.2    η=0.0

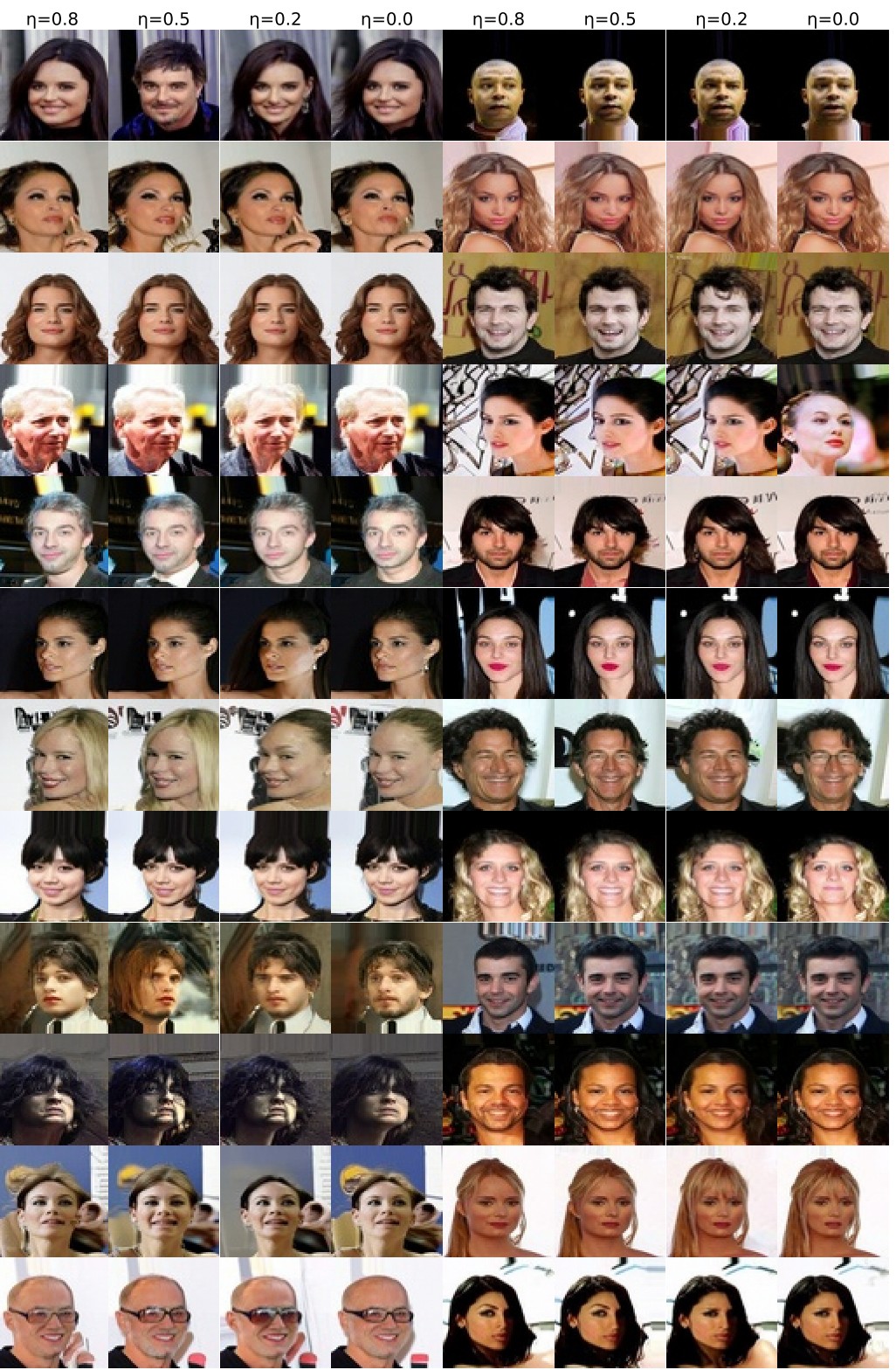

Figure 10: More samples generated by models trained on CelebA-50K with different mask ratios.

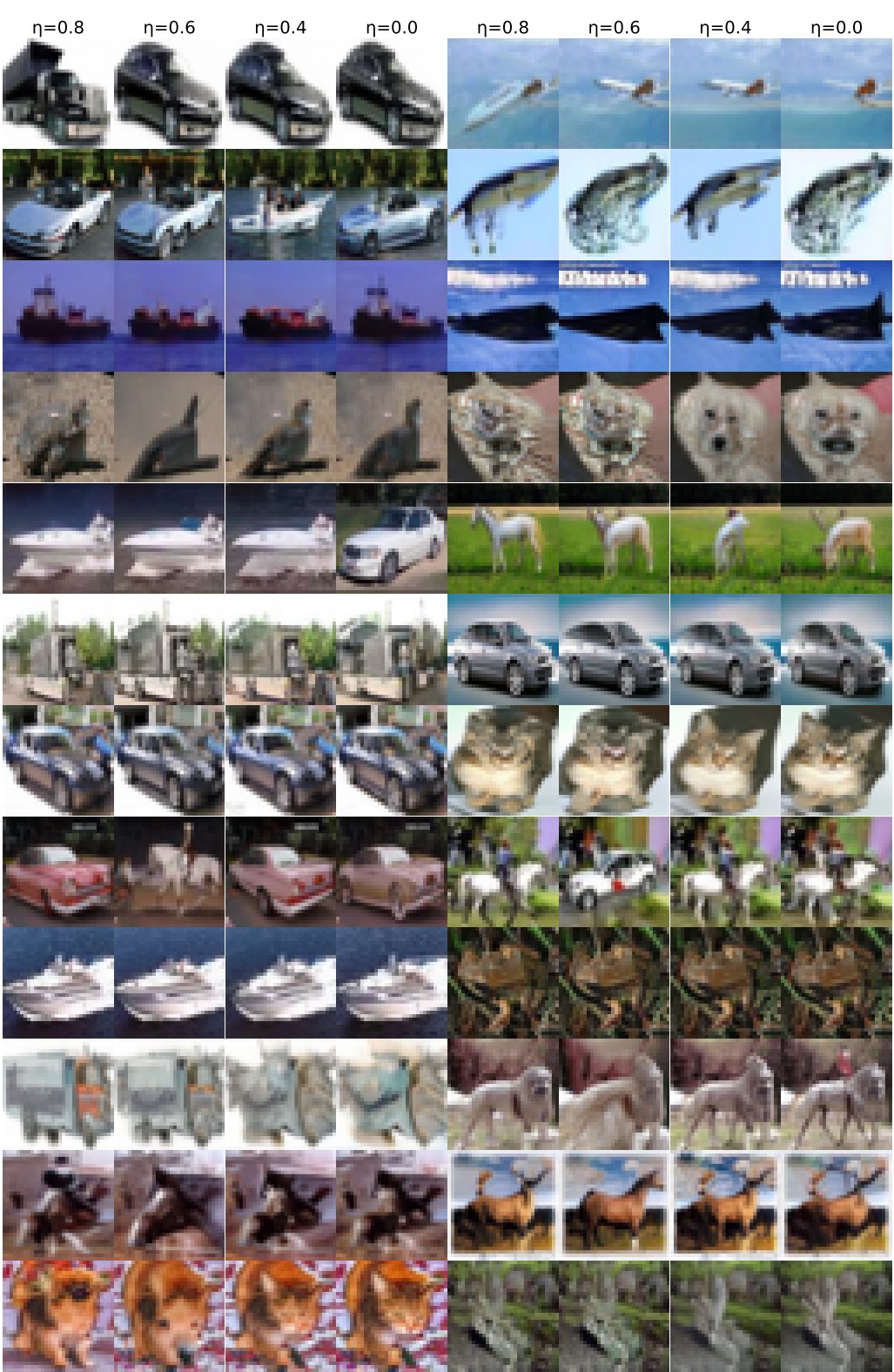

Figure 11: More samples generated by models trained on CIFAR10 with different mask ratios.

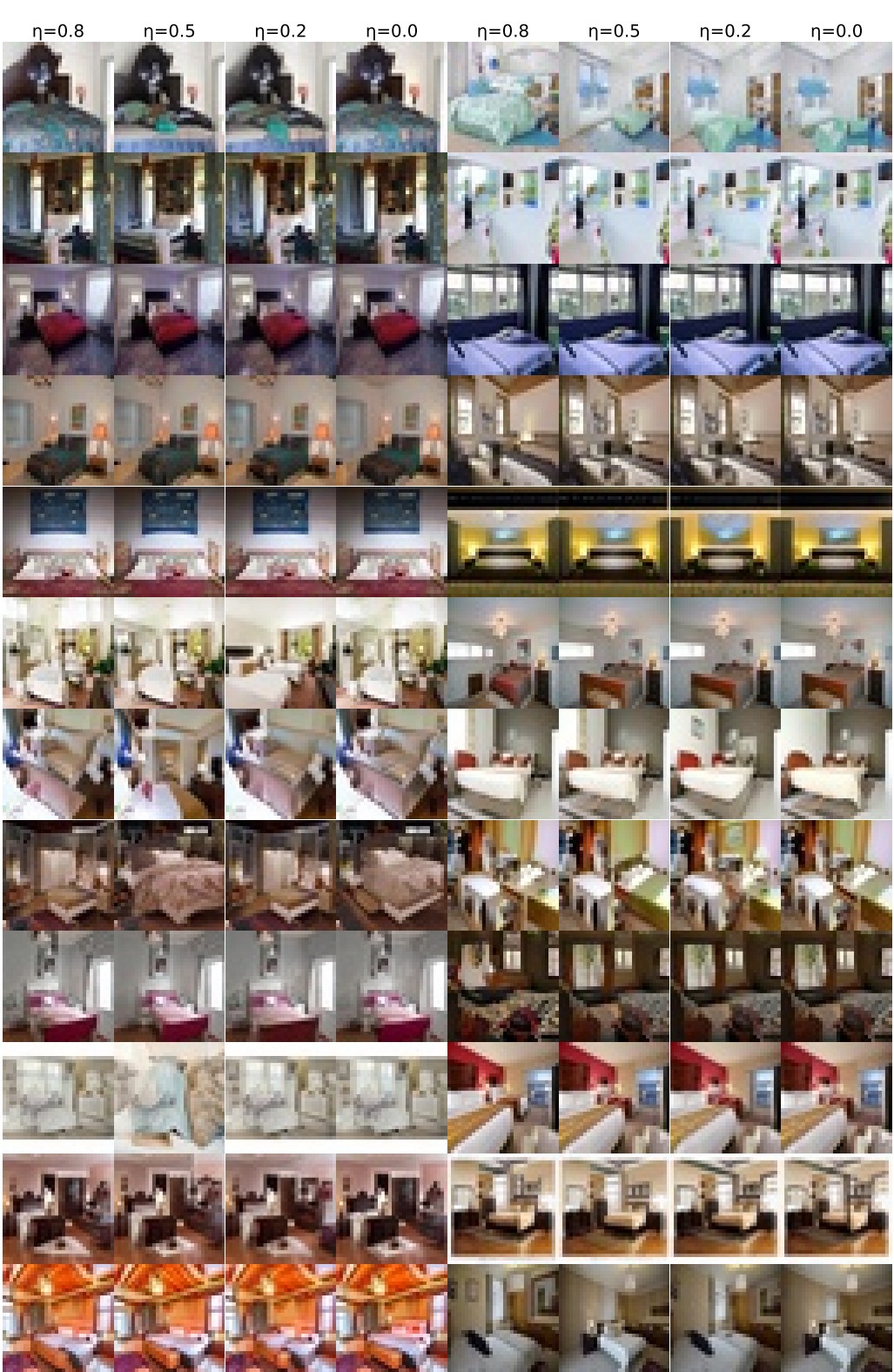

Figure 12: More samples generated by models trained on LSUN Bedroom with different mask ratios.

# E  DISCLOSURE OF LLMs USAGE

The draft was written by authors without using Large Language Models (LLMs). The ideas were formalized independently of LLMs assistance. LLMs were used to polish the draft, including assisting with word choice and improving grammar. The polished text was subsequently revised by the authors.

# F  MORE RESULTS

Table 3: Averaged FID ($\sigma$) for different datasets. Lower ($\downarrow$) is better.

| Dataset | FM | SMD (ours) |
|---|---|---|
| CIFAR10 | 2.28 (0.04) | **2.24 (0.05)** |
| LSUN | 1.40 (0.04) | **1.39 (0.01)** |
| ImageNet | **2.09 (0.07)** | 2.20 (0.03) |
| CelebA-50K | 2.23 (0.23) | **1.85 (0.14)** |
| CelebA-10K | 13.72 (-) | **5.10 (-)** |

Table 4: Comparison of empirical score estimation errors on a 2D Gaussian data distribution. Lower ($\downarrow$) is better.

| Metric | W/O Masking | W/ Masking (ours) |
|---|---|---|
| Mean | 3.65 | **2.78** |
| Max | 10.63 | **6.18** |

Table 5: Comparison of gradient sensitivity over 10,000 images on CIFAR10. Higher ($\uparrow$) is better.

| Metric | FM | SMD (ours) |
|---|---|---|
| Mean | 2055.16 | **2617.91** |
| Median | 2054.55 | **2617.16** |

Table 6: Averaged L2 distance to nearest training samples over 10,000 images on CelebA-10K. Higher ($\uparrow$) is better.

| Metric | FM | SMD (ours) |
|---|---|---|
| L2 Distance ($\uparrow$) | 42.32 (7.85) | **46.02 (8.03)** |

