# OpenReview forum: "Atrous Learning for Diffusion Models"
_ICLR.cc/2026/Conference — Submitted to ICLR 2026_

### Official Review · Reviewer_UQdc · 2025-10-18

**Soundness:** 2
**Presentation:** 2
**Contribution:** 2
**Rating:** 2
**Confidence:** 4

**Summary:**

The authors propose "atrous learning for diffusion models", where they introduce masking in the denoising objective in an effort to avoid "spatial inconsistencies". They claim that their approach mitigates undesirable effects of locality biases that are inherent in diffusion models. Experiments are conducted on synthetic data and standard image datasets, where improved performance is observed compared to baselines.

**Strengths:**

1. The paper is easy to read and the overall methodology is simple and clear.
2. Along with the experiments, the authors offer some theoretical discussion and attempt to connect their work with existing literature.
3. The authors have taken care to include details such as masking ratio ablations and performance as a function of the training epochs.
4. I found the angle explored in Section 5.4, regarding memorization, particularly interesting. I encourage the authors to explore this further.

**Weaknesses:**

1. While the methodology is clear to me, I am not convinced about the motivation of this work. The connection with prior work on Ω-locality is, in my opinion, loose. I include specific questions relating to this point in the Questions section below.

2. The demonstrations are not convincing. The experiment in Figure 3 is not particularly informative as I struggle to see differences in the gradient heatmaps. I fail to see any correlation between the mask ratio and the standard metrics reported in Tables 1, 2 and it is not clear whether the reported improvements are statistically significant. For the experiment in Figure 6, none of the the samples appear sufficiently close to the training data and the proposed method's effect is therefore unclear.

3. The paper advertises the proposed method as consistently outperforming baselines in the abstract. I feel that this framing is not appropriate as the reported improvements are questionable. Larger scale (e.g. ImageNet) and more thorough benchmarking with competitors would be required to properly validate such claims. The angle explored in Section 5.4, i.e., mitigating memorization, is, in my opinion, a better fit for this work (e.g., see [1] for similar analysis on language models).

**Questions:**

1. As I understand, the binary masks do not impose any locality constraints, i.e., they are iid. How is it possible then that SMD mitigates locality bias? Increased variance, as mentioned in Proposition 1, is not sufficient to force the networks to explore non-local structures. For example, one can also decrease the training batch size to achieve higher variance but there is no expectation that this mitigates locality.

2. I fail to see any evidence for the claims made in Section 6 (starting at line 471). For example, how might SMD promote "large-scale structure first and filling in finer detail later"?

3. Beyond the samples of Figure 2, could the authors provide more examples of locality bias and how this might be mitigated by SMD? At present, I am not convinced with the visualization on this toy dataset.

[Minor] Figure 1: Supervised Singals -> Supervised Signals

[1] Be like a Goldfish, Don't Memorize! Mitigating Memorization in Generative LLMs, NeurIPS 2024

---

> ### Author Response · Authors · 2025-11-27
> **Response to Reviewer UQdc [1/2]**
>
> ### 1. Loose connection to Omega locality
> > While the methodology is clear to me, I am not convinced about the motivation of this work. The connection with prior work on Ω-locality is, in my opinion, loose.
>
> We thank the reviewer for the valuable feedback. We have updated our paper accordingly and weaken the connection to $\Omega-$ locality, as it serves only as an inspiration for this work. Instead, we put more attention on the practical effect of the proposed masking strategy, as shown in the detailed reponse to your 2nd comment.
>
> ### 2. Weak performance
> > The demonstrations are not convincing. The experiment in Figure 3 is not particularly informative as I struggle to see differences in the gradient heatmaps. I fail to see any correlation between the mask ratio and the standard metrics reported in Tables 1, 2 and it is not clear whether the reported improvements are statistically significant. For the experiment in Figure 6, none of the the samples appear sufficiently close to the training data and the proposed method's effect is therefore unclear.
>
> We thank the reviewer for the insightful comments. We conducted additional experiments as detailed below. As shown in Figure 5 (updated), with more training runs, our method achieves FID scores comparable to the baseline model. Moreover, our experiments in Figures 5 (updated) and 7 (updated) demonstrate that masking helps the model leverage broader contextual information. Results in Figure 8 show that masking can mitigate training instability, and Table 1 indicates that it can reduce memorization. Overall, our findings suggest that masking is safe to use while providing clear benefits for diffusion model training.
>
> - **Large-scale and multipel runs experiments** We have added new experiments on ImageNet, which contains approximately 1.2M images. In addition, we conduct four training runs for each model across all four datasets: CIFAR-10, CelebA-50K, LSUN Bedroom, and ImageNet. As shown in **Figure 5** (updated paper), our method with 80\% masking achieves FID scores comparable to the baseline. Moreover, when training longer on CelebA-50K, the baseline model diverges, whereas our method remains stable.
> - **Quantified broader contextual representation** To examine the contextual representation induced by SMD, we complement the gradient heatmap visualization in Figure 3 with a quantitative analysis of gradient magnitudes over 10,000 images. As shown in **Figure 4** (updated paper), masking leads to significantly larger gradients over non-target pixel positions, demonstrating that SMD enables the model to leverage broader contextual information during pixel generation.
> - **Lower score approximation error** Since FID does not directly measure the accuracy of the learned score function, the true target in diffusion training, we conduct an additional 2D Gaussian experiment. As shown in **Figure 7** (updated paper), our method with masking, which effectively creates multiple partial views of limited data points, estimates the score function more accurately than the baseline.
> - **Preventing training divergence** We further show that diffusion models can suffer from training divergence when trained for long time. In **Figure 8** (updated paper), the results show that the baseline FID clearly goes up while SMD with up to 98\% masking can remain stable and even get lower FID.
> - **Mitigating memorization** We also quantitatively compare memorization between our method and the baseline. **Table 1**(updated paper) shows that on CelebA-10K, images generated by SMD have a larger L2 distance to the nearest training samples than those from the baseline, while achieving lower FID (**Figure 8** (updated paper)).

---

> > ### Author Response · Authors · 2025-11-27
> > **Response to Reviewer UQdc [2/2]**
> >
> > ### 3. Overclaim on performance and evidence for the claim in discussion
> > > The paper advertises the proposed method as consistently outperforming baselines in the abstract. I feel that this framing is not appropriate as the reported improvements are questionable. Larger scale (e.g. ImageNet) and more thorough benchmarking with competitors would be required to properly validate such claims. The angle explored in Section 5.4, i.e., mitigating memorization, is, in my opinion, a better fit for this work (e.g., see [1] for similar analysis on language models).
> >
> > > I fail to see any evidence for the claims made in Section 6 (starting at line 471). For example, how might SMD promote "large-scale structure first and filling in finer detail later"?
> >
> > We sincerely thank the reviewer for raising these points. We have updated our paper to rephrase the abstract (see below) and removed the corresponding claim from the discussion. Instead, we now focus on extensively examining the empirical benefits of the proposed masked training.
> >
> > > Experiments show that our method achieves competitive FIDs across datasets and prevents training from exploding when dataset is small. Moreover, the masking strategy helps mitigate memorization in diffusion models.
> >
> > ### 4. Not conviced on SMD mitigating locality bias and More examples of locality bias
> > >As I understand, the binary masks do not impose any locality constraints, i.e., they are iid. How is it possible then that SMD mitigates locality bias? Increased variance, as mentioned in Proposition 1, is not sufficient to force the networks to explore non-local structures. For example, one can also decrease the training batch size to achieve higher variance but there is no expectation that this mitigates locality.
> > > Beyond the samples of Figure 2, could the authors provide more examples of locality bias and how this might be mitigated by SMD? At present, I am not convinced with the visualization on this toy dataset.
> >
> > We thank the reviewer for the insightful feedback. Our method differs from using smaller batch sizes, which has been extensively studied as a hyperparameter tuning strategy. In SMD, the model is not trained on complete images. While limited by analysis tools, we can demonstrate the presence of unbiased gradients and increased variance. Importantly, masking also provides additional benefits, as evidenced in our experiments.
> >
> > We conducted new experiments to quantify how SMD mitigates locality and promotes broader contextual use during pixel generation. **Figure 4** (updated) shows the distribution of the L1 norm of gradients over 10,000 images, where SMD exhibits higher gradient sensitivity to non-target pixel positions.
> >
> > In **Figure 7** (updated), using a 2D Gaussian dataset where the ground-truth score is analytically known, we show that masking enables the model to estimate the score over a broader range, including regions far from the data points, resulting in lower approximation error compared to the baseline. We hope these new experiments help address your concerns.

---

### Official Review · Reviewer_J3hx · 2025-10-29

**Soundness:** 3
**Presentation:** 3
**Contribution:** 2
**Rating:** 2
**Confidence:** 3

**Summary:**

The authors point out that recent work on diffusion/flow models where it has been shown that for a given coordinate in the output space, information from a local neighborhood is mainly used to score or predict at the coordinate. They hypothesize that this *locality* of the diffusion process harms the model performance and that more global representations can be learned. To encourage such behaviour, they propose masking the training data such that the models are forced to consider more spread out information. Qualitative and quantitative analysis is done on outputs from models trained with the masked approach (SMD), suggesting that masking does in fact help.

**Strengths:**

The paper is well motivated, and the theory is mostly easy to follow. The idea that encouraging more global use of information is promissing (and you could maybe have referenced other work encouraging such behaviour, e.g. dino). The flow is good and the theoretical analysis of the SMD gradient is nice.

**Weaknesses:**

The main weakness of the paper is the empirical validation. The datasets used are small and have low resolution, qualitative analysis needs to be more thorough, and quantitative analysis is lacking. The FID table is alright (50k images is good!) but still only compares a single model per setting, right?

More importantly, the main hypothesis that global information is not being used without masking is also not tested well enough. One could imagine masking having a different role, such as being a regularizer.

Finally, there are no statistics or error bars, for this to be a convincing study you need to train multiple models. Pointwise evaluations of FID scores are not enough. This is also seen in figure 4 where the curves are not exactly smooth. Or in table 2 where it is not clear whether the fluctuations in the numbers are due to noise.

Because of this, I can't recommend acceptance (empirical evidence for the local/global claim and some statistics are needed, happy to reconsider if shown some)


More comments:

I found figure 1 slightly confusing, i recommend explaining the variables and labels in the caption (also a typo in "Singals"). It is not clear how the multiple masks for SMD are used (the PSPC dynamics seem somewhat illustrated with the colors, maybe something similar can be done).

In section 5.1 on spatial consistency, it would be nice to see some kind of quantitative evaluation rather than just 9 samples from each setup.

In section 5.2. on contextual representations, the single outputs are nice but again, some kind of quantitative result would be nice, and more examples.

Figure 3 needs much larger text, and the plots are barely different to the untrained eye, especially for the same timesteps, maybe just crop around the center?

Figre 4 and 5 are also way too small. and the qualitative results unclear.

**Questions:**

I am somwhat confused by the statement in section 5.2 that says that the same noise provided to different models leads to two different images. Why shouldn't it? (the gradient difference could be visualized/quantified better)

Can you think of a concrete test for whether a model has learned to use global information instead of local information to predict?

How exactly are the recall and precision in Table 2 defined/calculated?

Small nit, why do you have different masking ratios in 4a than in 4b and 4c?

---

> ### Author Response · Authors · 2025-11-27
> **Response to Reviewer J3hx [1/2]**
>
> ### 1. More runs for each model
> > The main weakness of the paper is the empirical validation. The datasets used are small and have low resolution, qualitative analysis needs to be more thorough, and quantitative analysis is lacking. The FID table is alright (50k images is good!) but still only compares a single model per setting, right?
>
> > Finally, there are no statistics or error bars, for this to be a convincing study you need to train multiple models. Pointwise evaluations of FID scores are not enough. This is also seen in figure 4 where the curves are not exactly smooth. Or in table 2 where it is not clear whether the fluctuations in the numbers are due to noise.
>
> We thank the reviewer for the valuable feedback on the experiment design. We conduct **four** training runs for each model across all four datasets: CIFAR-10, CelebA-50K, LSUN Bedroom, and **ImageNet (newly added)**. As shown in Figure 5 (updated paper), our method with 80% masking achieves FID scores comparable to the baseline. Moreover, when training longer on CelebA-50K, the baseline model diverges, whereas our method remains stable.
>
> ### 2. The global information in baseline and the role of SMD as a regularizer
> > More importantly, the main hypothesis that global information is not being used without masking is also not tested well enough. One could imagine masking having a different role, such as being a regularizer.
>
> Thanks for your insightful comments and agree with you that although locality have been shown one mechanism in diffusion models, the global information can also be used. We would like to clarify that we argue that masking can further improve the global information use in diffusion models. We conduct experiments to quantify the contextual representations in **Figure 4** (updated paper), where our method with masking shows clearly larger L1 norm of the gradients w.r.t. all the image pixels. Furthermore, the 2D Gaussian experiment in **Figure 7** (updated paper) can also show that the Empirical score estimated with masking has broader accuracy then the baseline which can only estimate the scores near the given data points.
>
> ### 3. Figure texts are too small
> We appreciate your feedback and apologize for the uncomfortable presentation. We have updated figures. We hope the current version can make it easier to read.
>
> ### 4. Quantity on spatial consistency and contextual representations
> > In section 5.1 on spatial consistency, it would be nice to see some kind of quantitative evaluation rather than just 9 samples from each setup. In section 5.2. on contextual representations, the single outputs are nice but again, some kind of quantitative result would be nice, and more examples.
>
> Thanks for the valuable feedback. We have updated experiments and added more quantative comparisons. The new results in **Figure 4** (updated paper) and **Figure 7** (updated paper) demonstrate that the proposed masking can help account for broader contextual representations.

---

> > ### Author Response · Authors · 2025-11-27
> > **Response to Reviewer J3hx [2/2]**
> >
> > ### 5. Same noise from different models leads to two different images. Why shouldn't it?
> > > I am somwhat confused by the statement in section 5.2 that says that the same noise provided to different models leads to two different images. Why shouldn't it? (the gradient difference could be visualized/quantified better)
> >
> > We thank the reviewer for this detailed observation. As shown in both existing works [1] and our experiments (Figure 6), diffusion models trained with different architectures, losses, or even subsets of datasets tend to generate similar images when given the same noise. However, by applying masking that encourages broader contextual influence for each pixel, our model can sometimes generate images that differ from the baseline even when using the same noise input.
> >
> > [1] Kadkhodaie, Zahra, et al. "Generalization in diffusion models arises from geometry-adaptive harmonic representations." The Twelfth International Conference on Learning Representations.
> >
> > ### 6. A concrete test for global information use
> > > Can you think of a concrete test for whether a model has learned to use global information instead of local information to predict?
> >
> > In **Figure 4** (updated paper), we quantify the L1 norm of gradients with respect to all pixels. The results show that our method exhibits higher gradient sensitivity to other pixels, indicating that it leverages more global contextual information.
> >
> > In **Figure 7** (updated paper), we present a numerical experiment on a 2D Gaussian data distribution. The score estimated with masking, using the same 10 data points, captures a broader range of the distribution, whereas the baseline fails to account for regions without data points.
> >
> > ### 7. Recall and precision computation
> > > How exactly are the recall and precision in Table 2 defined/calculated?
> >
> > We apologize for the unclear explanation. We follow the computation in [2] to calculate recall and precision. Note that we have updated our experiments to account for the training divergence issue observed in diffusion models trained on CelebA-10K, and we have extensively evaluated generation metrics on large-scale datasets.
> >
> > [2] Kynkäänniemi, Tuomas, et al. "Improved precision and recall metric for assessing generative models." Advances in neural information processing systems 32 (2019).
> >
> >
> > ### 8. Why different masking ratios?
> > > why do you have different masking ratios in 4a than in 4b and 4c?
> >
> > We apologize for the confusion. Initially, we set masking ratios in the range $[0,1)$. Due to computational limitations, we examined fine-grained masking ratios only on a single dataset, CIFAR-10. Our experiments indicate that it is safe to use masking ratios as high as 98\%, as shown in Figure 8. Nonetheless, more sophisticated designs for choosing masking ratios could be explored in future work.

---

### Official Review · Reviewer_zgz8 · 2025-11-01

**Soundness:** 3
**Presentation:** 3
**Contribution:** 2
**Rating:** 4
**Confidence:** 3

**Summary:**

This submission proposes a new diffusion model to mitigate spatial inconsistency caused by the models' locality.
The proposed method utilizes atrous (dilated) convolution strategy combined with Simplified Masked Diffusion (SMD) training.
Compared with the original diffusion models, SMD provided training gradients with higher variance without bias.
Experiments are conducted using some toy data and CIFAR10 32×32, CelebA-50K 64×64, and LSUN Bedroom 32×32. Improved FIDs and precision and recall of generated images are reported.

**Strengths:**

- Masking strategy is . Deeply investigating the effect of masking in convolutional diffusion models is a meaningful direction of research.
- Mitigating locality dependency seems a promising direction, especially non-attentive convolutional diffusion models.

**Weaknesses:**

- The organization/technical clarity of the paper is not excellent. "Atrous learning" is featured in the title and Introduction, but the relationship between atrous learning and SMD is not clear. Details of the network architecture and other training parameters are missing.

- Omega locality is defined in Sec 4.2 using a certain amount of text but is not used in the following theoretical and empirical analyses, which causes a feeling of somewhat shallow discussion.

- Experiments are limited in the small-scale datasets, and the generality of the method for larger scales are unclear.

**Questions:**

- Which network architecture was used and how atrous convolution was incorporated? Or atrous convolution is not used but the masked training is termed "à trou" (holed)?

---

> ### Author Response · Authors · 2025-11-27
> **Response to Reviewer zgz8**
>
> ### 1. Clarity of the used network
> >The organization/technical clarity of the paper is not excellent. "Atrous learning" is featured in the title and Introduction, but the relationship between atrous learning and SMD is not clear. Details of the network architecture and other training parameters are missing.
>
> We are sorry for the uncleared explaination. We have updated the paper in the introduction as follows:
> >The name is inspired by the \textit{atrous convolution} operation~\citep{chen2017rethinking}, but instead of introducing holes in convolution kernels, we introduce them in the training losses.
>
> Moreover, we clarify that we use U-Net for both our method and the baselien across all experiments. We thank the reviewer for pointing this out.
>
> ### 2. How Omega locality is connected to SMD
> >Omega locality is defined in Sec 4.2 using a certain amount of text but is not used in the following theoretical and empirical analyses, which causes a feeling of somewhat shallow discussion.
>
> We thank the reviewer for pointing out this issue. We have updated the paper and reduced the emphasis on the $\Omega$-locality, as it serves only as an inspiration for this work. Instead, we put more attention on the practical effect of the proposed masking strategy, as shown in the detailed reponse to your last comment.
>
> ### 3. Large-scale experiments
> >Experiments are limited in the small-scale datasets, and the generality of the method for larger scales are unclear.
>
> We greatly appreciate the reviewer’s valuable feedback. We have now updated the experimental section accordingly.
> - **Large-scale and multipel runs experiments** We have added new experiments on ImageNet, which contains approximately 1.2M images. In addition, we conduct four training runs for each model across all four datasets: CIFAR-10, CelebA-50K, LSUN Bedroom, and ImageNet. As shown in **Figure 5** (updated paper), our method with 80% masking achieves FID scores comparable to the baseline. Moreover, when training longer on CelebA-50K, the baseline model diverges, whereas our method remains stable.
> - **Quantified broader contextual representation** To examine the contextual representation induced by SMD, we complement the gradient heatmap visualization in Figure 3 with a quantitative analysis of gradient magnitudes over 10,000 images. As shown in **Figure 4** (updated paper), masking leads to significantly larger gradients over non-target pixel positions, demonstrating that SMD enables the model to leverage broader contextual information during pixel generation.
> - **Lower score approximation error** Since FID does not directly measure the accuracy of the learned score function, the true target in diffusion training, we conduct an additional 2D Gaussian experiment. As shown in **Figure 7** (updated paper), our method with masking, which effectively creates multiple partial views of limited data points, estimates the score function more accurately than the baseline.
> - **Preventing training divergence** We further show that diffusion models can suffer from training divergence when trained for long time. In **Figure 8** (updated paper), the results show that the baseline FID clearly goes up while SMD with up to 98% masking can remain stable and even get lower FID.
> - **Mitigating memorization** We also quantitatively compare memorization between our method and the baseline. **Table 1**(updated paper) shows that on CelebA-10K, images generated by SMD have a larger L2 distance to the nearest training samples than those from the baseline, while achieving lower FID (**Figure 8** (updated paper)).
>
> Through the extensive experiments, we show that SMD offers a *free lunch* for diffusion model training with several advantages: (1) It achieves comparable FID scores across datasets while avoiding long-training instability. (2) It improves underlying population score estimation. (3) SMD can mitigate the memorization problem in diffusion models. (4) Remarkably, SMD maintains comparable performance even when up to 98% of pixels are masked, suggesting valuable implications for understanding the dynamics of diffusion models.

---

> > ### Comment · Reviewer_zgz8 · 2025-11-28
> > **Post rebuttal comment**
> >
> > I appreciate the Authors' response.
> > Some of my concerns about technical clarity are answered. However, the added results in Fig. 5 and Fig. 6 seem to show somewhat limited improvements. I would keep my initial rating.

---

> ### Author Response · Authors · 2025-11-28
> **Further Response to Reviewer zgz8**
>
> We thank the reviewer for your prompt response and we are happy that we have already partially solved your concerns, especially on technical clarity. For your concerns regarding the performance improvement, we further report the following numbers as a clear evalution of our method. Note that **we are not only focusing on the FID scores, but also concerning other important problems in diffusion models**. For example, the instability of long training, memorization in generation (**as suggested by Reviewer UQdc**), and unrealistic generation (unaccurate population score estimation). We conduct comprehensive comparison on these important metrics, and we believe the results in the following confirm the benefits brought by the proposed masking strategy.
>
> Table 1 (Generation quality): Averaged FID (σ) for different datasets. Lower (↓) is better.
>
> | Dataset    | FM              | SMD (ours)      |
> | ---------- | --------------- | --------------- |
> | CIFAR10    | 2.28 (0.04)     | **2.24 (0.05)** |
> | LSUN       | 1.40 (0.04)     | **1.39 (0.01)** |
> | ImageNet   | **2.09 (0.07)** | 2.20 (0.03)     |
> | CelebA-50K | 2.23 (0.23)     | **1.85 (0.14)** |
> | CelebA-10K | 13.72 (-)       | **5.10 (-)**    |
>
> Table 2 (Memorization): Averaged L2 distance to nearest training samples over 10,000 images. Higher (↑) is better.
> | Metric          | FM           | SMD (ours)       |
> | --------------- | ------------ | ---------------- |
> | L2 Distance (↑) | 42.32 (7.85) | **46.02 (8.03)** |
>
>
> Table 3 (Score estimation error): Comparison of empirical score estimation errors. Lower (↓) is better.
> | Metric | W/O Masking | W/ Masking (ours) |
> | ------ | ----------- | ----------------- |
> | Mean   | 3.65        | **2.78**          |
> | Max    | 10.63       | **6.18**          |
>
> Table 4 (Contextual information use): Comparison of gradient sensitivity over 10,000 images. Higher (↑) is better.
> | Metric | FM      | SMD (ours)  |
> | ------ | ------- | ----------- |
> | Mean   | 2055.16 | **2617.91** |
> | Median | 2054.55 | **2617.16** |

---

### Author Response · Authors · 2025-11-27
**General response to all reviewers**

We thank all the reviewers for their valuable time and feedback. We are pleased that reviewers appreciate our idea and consider the proposed masked training a **promising direction** (Reviewer zgz8, Reviewer J3hx). Reviewer UQdc also finds the method particularly **interesting** with respect to memorization. At the same time, we acknowledge reviewers’ concerns regarding our experimental evaluations and the logical connection to locality. To address these points, we have conducted additional experiments and updated our paper accordingly. Specifically, we have revised and expanded the following experiments:

1. **Large-scale and multipel runs experiments** We have added new experiments on ImageNet, which contains approximately 1.2M images. In addition, we conduct **four training runs** for each model across all four datasets: **CIFAR-10, CelebA-50K, LSUN Bedroom, and ImageNet**. As shown in **Figure 5** (updated paper), our method with 80% masking achieves FID scores comparable to the baseline. Moreover, when training longer on CelebA-50K, the baseline model diverges, whereas our method remains stable.
2.  **Quantified broader contextual representation** To examine the contextual representation induced by SMD, we complement the gradient heatmap visualization in Figure 3 with a quantitative analysis of gradient magnitudes over 10,000 images. As shown in **Figure 4** (updated paper), masking leads to significantly larger gradients over non-target pixel positions, demonstrating that SMD enables the model to leverage broader contextual information during pixel generation.
3. **Lower score approximation error** Since FID does not directly measure the accuracy of the learned score function, the true target in diffusion training, we conduct an additional 2D Gaussian experiment, where we can analytically compute score functions. As shown in **Figure 7** (updated paper), our method with masking, which effectively creates multiple partial views of limited data points, estimates the score function more accurately than the baseline (5th subfigure from left to right) and captures a broader range beyond the observed data points (3rd subfigure), closely matching the ground-truth population score (1st subfigure).
4. **Preventing training divergence** We further show that diffusion models can suffer from training divergence when trained for long time. In **Figure 8** (updated paper), the results show that the baseline FID clearly goes up while **SMD with up to 98\% masking can remain stable and even get lower FID**.
5. **Mitigating memorization** We also quantitatively compare memorization between our method and the baseline. **Table 1**(updated paper) shows that on CelebA-10K, images generated by SMD have a larger L2 distance to the nearest training samples than those from the baseline, while achieving lower FID (**Figure 8** (updated paper)).

Through the extensive experiments, we show that SMD offers a *free lunch* for diffusion model training with several advantages:
1. It achieves comparable FID scores across datasets while avoiding long-training instability.
3. It improves underlying population score estimation.
4. SMD can mitigate the memorization problem in diffusion models.
5. Remarkably, SMD maintains comparable performance even when up to 98\% of pixels are masked, suggesting valuable implications for understanding the learning dynamics of diffusion models.

We hope the updated paper with new experiments could help address the reviewers' concerns. We look forward to your further feedback.

---

> ### Author Response · Authors · 2025-11-28
> **General response to all reviewers [2]**
>
> We thank all reviewers for their valuable comments. Here, we report detailed comparison results between our method, SMD, and the baseline flow matching (FM) across several benchmarks: (1) FID scores on five datasets, (2) L2 distances to training samples over 10,000 generated images on CelebA-10K, (3) empirical score estimation errors on a 2D Gaussian dataset, and (4) gradient sensitivity measured over 10,000 generated images on CIFAR-10. These comparisons offer a comprehensive evaluation of our method:
> - Our method demonstrates the ability to **improve generative quality**.
> - It effectively **prevents the training instability problem** when diffusion models trained for long time.
> - It **mitigates the memorization problem** in diffusion model generation.
> - It **accurately estimates population score** functions, thereby preventing unrealistic generation.
> - It effectively **leverages contextual information** during pixel generation.
>
> Table 1 (Generation quality): Averaged FID (σ) for different datasets. Lower (↓) is better.
>
> | Dataset    | FM              | SMD (ours)      |
> | ---------- | --------------- | --------------- |
> | CIFAR10    | 2.28 (0.04)     | **2.24 (0.05)** |
> | LSUN       | 1.40 (0.04)     | **1.39 (0.01)** |
> | ImageNet   | **2.09 (0.07)** | 2.20 (0.03)     |
> | CelebA-50K | 2.23 (0.23)     | **1.85 (0.14)** |
> | CelebA-10K | 13.72 (-)       | **5.10 (-)**    |
>
> Table 2 (Memorization): Averaged L2 distance to nearest training samples over 10,000 images. Higher (↑) is better.
> | Metric          | FM           | SMD (ours)       |
> | --------------- | ------------ | ---------------- |
> | L2 Distance (↑) | 42.32 (7.85) | **46.02 (8.03)** |
>
>
> Table 3 (Score estimation error): Comparison of empirical score estimation errors. Lower (↓) is better.
> | Metric | W/O Masking | W/ Masking (ours) |
> | ------ | ----------- | ----------------- |
> | Mean   | 3.65        | **2.78**          |
> | Max    | 10.63       | **6.18**          |
>
> Table 4 (Contextual information use): Comparison of gradient sensitivity over 10,000 images. Higher (↑) is better.
> | Metric | FM      | SMD (ours)  |
> | ------ | ------- | ----------- |
> | Mean   | 2055.16 | **2617.91** |
> | Median | 2054.55 | **2617.16** |

---

### Author Response · Authors · 2025-12-03
**Paper Summary for AC**

We sincerely thank the area chair and all reviewers for the time and constructive feedback dedicated to our work. We have substantially improved the paper, and the new experiments further confirm several advantages of the proposed method, as summarized below.

### Main Idea
Aiming for the critical issue of spatial inconsistency in diffusion models, we propose atrous learning, a simple yet effective masking strategy (SMD) that can be implemented with only a few lines of code. Specifically, during the computation of the regression loss, we randomly mask pixel locations within the training images. The key insight is that such masking exposes the model to only partial supervisory signals, thereby compelling it to rely on broader contextual information when generating samples. This encourages more coherent spatial structures and mitigates inconsistency artifacts.

### Experimental Results
Our extensive experiments show the following results (please check this [anonymous website](https://sites.google.com/view/atrous-learning/home)):
- The proposed SMD achieves **competitive FID scores** across 5 datasests, even when up to 98% of pixels are masked. See [**Figure 5**](https://sites.google.com/view/atrous-learning/home#h.bmzjejz7t6k9) in the updated paper or the link to the anonymous website.
- SMD **prevents the training instability** problem when diffusion models trained for long time, as shown in [**Figure 8**](https://sites.google.com/view/atrous-learning/home#h.qqga6olh68r3).
- It effectively **leverages broader contextual information** during pixel generation, as shown in [**Figure 4**](https://sites.google.com/view/atrous-learning/home#h.8r8pfecw58r6).
- It **mitigates the memorization** problem in diffusion model generation, as shown in [**Table 1**](https://sites.google.com/view/atrous-learning/home#h.zbsjlaolp27u).
- It **accurately estimates population score functions** on a 2D Gaussian distribution example, as shown in [**Figure 7**](https://sites.google.com/view/atrous-learning/home#h.mupbryhql8m).
- On a toy dataset, we show **improved spatial consistency** by qualitatively visualizing the generated samples, as shown in [**Figure 2**](https://sites.google.com/view/atrous-learning/home#h.m9szgdh1h3ln).

Overall, we believe that the diverse experiments substantiate the benefits of the proposed SMD. Furthermore, the findings obtained under high masking ratios offer valuable insights for the diffusion community on understanding the training dynamics of diffusion models.

---

> ### Author Response · Authors · 2025-12-03
> **Review Summary for AC**
>
> We thank the reviwers for providing valuable feedback to our work, which has substantially improved our paper.
>
> ### Strength 1: Interesting idea and promising direction
> We are glad to see that all reviewers like our idea and think the proposed method is promising:
>
> `Reviewer zgz8:` Meaningful direction of research and a promising direction;
>
> `Reviewer J3hx:` The idea that encouraging more global use of information is promissing;
>
> `Reviewer UQdc:` I found the angle regarding memorization particularly interesting.
>
> However, we found that the reviewers’ primary concerns centered on the empirical evaluation. To address these issues, we have conducted extensive new experiments.
> ### Main Concern 1: Experimental results on FID scores
>
> `Reviewer zgz8:` Experiments are limited in the small-scale datasets.
>
> `Reviewer J3hx:` The main weakness of the paper is the empirical validation. There are no statistics or error bars.
>
> `Reviewer UQdc:` The demonstrations are not convincing. It is not clear whether the reported improvements are statistically significant.
>
> **Response:** For the FID scores, we added new experiments on ImageNet and perform 4 runs for each setting. As shown in [**Figure 5**](https://sites.google.com/view/atrous-learning/home#h.bmzjejz7t6k9), our method achieves competitive FID scroses across 5 datasets. Notablely, our method with up to 98% masking works well and can further avoid the training explodation of the baseline method, as shown in [**Figure 8**](https://sites.google.com/view/atrous-learning/home#h.qqga6olh68r3).
>
> ### Main Concern 2: Evidence for use of global information and reducing memorization
> Since our initial paper only qualitatively showed the improved global information use and reducing memorization, some reviewers were not fully convinced.
>
> `Reviewer J3hx:` (1) The main hypothesis that global information is not being used without masking is also not tested well enough. (2) It would be nice to see some kind of quantitative evaluation spatial consistency and on contextual representations.
>
> `Reviewer UQdc:` The experiment in Figure 3 is not particularly informative as I struggle to see differences in the gradient heatmaps. The proposed masking is not sufficient to force the networks to explore non-local structures.
>
> **Response:** We added new experiments to **quantify** the improved global information use ([**Figure 4**](https://sites.google.com/view/atrous-learning/home#h.8r8pfecw58r6)), and reduced memorization ([**Table 1**](https://sites.google.com/view/atrous-learning/home#h.zbsjlaolp27u)), which confirm the benefits of the proposed masking strategy. Moreover, in a 2D Gaussian experiment where the ground-truth score functions can be computed analytically, our method achieves more accurate score estimation than the baseline, as shown in [**Figure 7**](https://sites.google.com/view/atrous-learning/home#h.mupbryhql8m).
>
> ### Other concerns: clarity issues
> `Reviewer zgz8:` The relationship between atrous learning and SMD is not clear. Connection to Omega locality.
>
> `Reviewer J3hx:` I recommend explaining the variables and labels in the caption of Figure 1. Figure 3 needs much larger text. Figre 4 and 5 are also way too small.
>
> `Reviewer UQdc:` Overclaimed performance on FID scores. I fail to see any evidence for the claims made in the Discussion section.
>
> **Response:** We have updated our paper to clearly explain the relation betwen atrous learning and SMD, as well as the connection to Omega locality which is presented as our motivated angle. We have reploted the figures. We have rephrased the claims on FID scores and removed the claims in discussion. Note that we are not only focusing on FID scores, but also concerning on other critical aspects of diffusion models such as limited contextual information use and memorization issues.
>
> ### Summary
> We appreciate that all reviewers found the proposed idea promising. We fully understand the concerns raised regarding the empirical evaluation in the initial submission. In response, we have conducted extensive new experiments, and the results substantiate the advantages of our method. We believe our findings can inspire the diffusion community to address the critical issue of spatial inconsistency in current diffusion models, a challenge that extends beyond merely pursuing improved FID scores.

---

### Meta-Review · Area_Chair_E4G9 · 2025-12-28

**Summary:**

This paper proposes a simple and clear training method for diffusion models, i.e., applying a masking strategy in the training loss.

Reviewers largely appreciated that the masking is a meaningful and straightforward strategy. It is well motivated by the intention to reduce locality bias and encourage more global information learning. Reviewers also commented positively on the writing and the discussion about the memorisation.

Reviewers have significant concerns about the limited and small-scale experiments, the unclear connection to omega locality theory, the unconvincing demonstrations of results, and some questionable claims.

Most importantly, the results are not significant enough, and the experiments include only one baseline. It is unclear whether the conclusion is widely applicable.

**Reviewer Concerns:**

(Weaknesses are indexed using reviewers' original ordering)

For reviewer zgz8, W1 and W2 have been addressed. Although with some extra evidence, W3 is still not fully addressed.

For reviewer J3hx, major W1, W2, and W3 have been addressed partially.

For reviewer UQdc, W1 has been addressed. W2 has not been well addressed. The writing aspect of W3 has been addressed, but the experimental aspect of W3 has not been well addressed.

**Reviewer Scores:**

For reviewer zgz8, the score is unlikely to be increased.

For reviewer J3hx, the score will be the same or slightly increased to 4.

For reviewer UQdc, the score is unlikely to be increased .

---

### Decision · Program_Chairs · 2026-01-26

Reject